# Systemic Therapy for Advanced Hepatocellular Carcinoma: Current Stand and Perspectives

**DOI:** 10.3390/cancers15061680

**Published:** 2023-03-09

**Authors:** Daniel M. Girardi, Lara P. Sousa, Thiago A. Miranda, Fernanda N. C. Haum, Gabriel C. B. Pereira, Allan A. L. Pereira

**Affiliations:** 1Hospital Sírio-Libanes, SGAS 613/614 Conjunto E Lote 95-Asa Sul, Brasília 70200-730, Brazil; 2Hospital de Base do Distrito Federal, SMHS-Área Especial, Q. 101-Asa Sul, Brasília 70330-150, Brazil; 3Escola Superior de Ciências em Saúde, SMHN Conjunto A Bloco 01 Edifício Fepecs-Asa Norte, Brasília 70710-907, Brazil

**Keywords:** hepatocellular carcinoma, immunotherapy, targeted therapy, biomarkers, molecular landscape

## Abstract

**Simple Summary:**

Hepatocellular carcinoma is an aggressive disease with a poor prognosis. Treatment options for advanced disease have changed substantially in the last few years with the development of targeted therapy, immunotherapy and combinations of both treatment options. This development has led to an increase in disease control and overall survival. The aim of this review article is to summarize the current treatment options and future perspectives on the treatment of advanced hepatocellular carcinoma.

**Abstract:**

Hepatocellular carcinoma often develops in the context of chronic liver disease. It is the sixth most frequently diagnosed cancer and the third most common cause of cancer-related mortality worldwide. Although the mainstay of therapy is surgical resection, most patients are not eligible because of liver dysfunction or tumor extent. Sorafenib was the first tyrosine kinase inhibitor that improved the overall survival of patients who failed to respond to local therapies or had advanced disease, and for many years, it was the only treatment approved for the first-line setting. However, in recent years, trials have demonstrated an improvement in survival with treatments based on immunotherapy and new targeting agents, thereby extending the treatment options. A phase III trial showed that a combination of immunotherapy and targeted therapy, including atezolizumab plus bevacizumab, improved survival in the first-line setting, and is now considered the new standard of care. Other agents and combinations are being tested, including the combination of nivolumab plus ipilimumab and tremelimumab plus durvalumab, and they reportedly have clinical benefits. The aim of this manuscript is to review the latest approved therapeutic options in first- and second-line settings for advanced HCC and discuss future perspectives.

## 1. Introduction

Hepatocellular carcinoma (HCC) is a primary malignant liver tumor and the third leading cause of cancer-related deaths worldwide [1]. It is a major health problem worldwide because of its poor prognosis and increasing incidence with advancing age in all populations, reaching a peak at 70 years old [2]. It usually develops in the context of chronic liver disease or cirrhosis due to alcohol use, aflatoxin exposure, chronic hepatitis B or C virus infections, nonalcohol-associated steatohepatitis, genetic hemochromatosis or alpha-1-antitrypsin deficiency [3,4]. Despite several screening and diagnosis mechanisms, HCC is frequently diagnosed late in its course, especially because of the absence of symptoms in patients with early disease [5]. Patients should be evaluated carefully for better treatment decisions. This evaluation not only includes appropriate cancer staging, but also the degree of liver dysfunction and the control of comorbidities, including cirrhosis and chronic hepatitis.

The Child-Pugh criteria are used to evaluate liver function with the ascites degree, albumin and bilirubin concentration in serum levels, prothrombin time and encephalopathy degree [6]. Another algorithm that is widely used for staging and treatment decisions is the Barcelona Clinic of Liver Cancer (BCLC) approach. The BCLC classification helps to guide the management of patients with HCC by providing a standardized way to assess the extent of the disease and predict the patient’s prognosis. It takes into account the size and number of tumors as well as the patient’s underlying liver function and performance status. This information is used to assign patients to one of five stages of disease, from the early stage (0 and A) to the intermediate (B) and advanced stages (C and D) [7]. The classification also provides recommendations for treatment options at each stage, which can include surgical resection, liver transplantation, ablation, intra-arterial therapies and systemic therapy as well as supportive care. This variety helps to optimize the treatment of the patient [8].

Currently, treatment options for advanced disease are more diversified than ever because of developments in molecular and immune-based therapies [8]. For a decade, sorafenib, a multikinase inhibitor that has reported activity against multiple pathways, such as Raf-1, B-Raf, vascular endothelial growth factor receptor (VEGFR) 2, platelet-derived growth factor receptor (PDGFR) and c-Kit receptors [9], was the only treatment approved in the first-line setting of advanced disease based on the results of two phase III trials: SHARP and Asia-Pacific. The SHARP trial assigned patients with advanced HCC who had not received previous systemic treatment to receive either sorafenib or a placebo. The results showed a median overall survival (OS) of 10.7 months in the sorafenib group and 7.9 months in the placebo group (HR: 0.69; *p* < 0.001). The Asia-Pacific trial also randomly assigned patients with advanced HCC to receive either sorafenib or a placebo. The median OS was 6.5 months in patients treated with sorafenib and 4.2 months in patients who received the placebo (HR: 0.68; *p* = 0.014) [10,11]. Over the last few years, the treatment options in the first-line setting have changed substantially with the arrival of immunotherapy and new targeting agents. The aim of this manuscript is to review the latest approved therapeutic options in first- and second-line settings for advanced HCC, by focusing on immunotherapy and targeted therapies.

## 2. Molecular Landscape

### 2.1. Carcinogenesis and Drivers

The vast majority of HCCs arise in cirrhotic livers or with chronic liver disease. HCC follows a sequence of stages that can be observed through histopathological examination. The process begins with the development of dysplastic nodules, which can ultimately turn into HCC [12]. This progression is caused by genetic mutations within regenerating hepatocytes, particularly in the presence of inflammation and fibrosis [13]. The accumulation of these genetic and epigenetic changes is crucial in the development of liver cancer. HCC tumors typically have 60–70 genetic mutations, with many being “passenger mutations” that do not directly contribute to carcinogenesis, but some mutations occur in “driver genes” that activate the pathways important for the development of liver cancer [14,15]. The most frequent alteration, which is observed in 20% of high-grade dysplastic lesions and up to 60% of early HCC, is the reactivation of telomerase reverse transcriptase (*TERT*) [16]. Other promoter mutations are frequently altered in HCC in pathways that are related to the following: cell cycle control (*TP53*, *CDKNA2* and *CCND1*); chromatin modifiers (*ARID1A* and *ARID2*); the RTK/KRAS/PI3K pathway (*RPS6KA3*, *PIK3CA*, *KRAS*, *NRAS*, *FGF19* and *VEGFA*); oxidative stress (*NFE2L2* and *KEAP1*); and the Wnt/β-catenin pathway (*CTNNB1* and *AXIN1*) [17,18]. Upon exploring the progression of HCC in cirrhotic and noncirrhotic livers, an increased number of gene mutations were identified, along with chromosome alterations and dysplastic micronodule malignant transformation to poor prognosis. TERT promoter mutations are frequent at early stages, but CTNNB1 and TP53 mutation frequencies increase with progression, and focal amplifications at the FGF-CCND1 locus are mostly present in HCCs with a poor prognosis [19].

### 2.2. Immunology of HCC

The liver performs an important role in metabolism by either excreting toxic waste substances or filtering environmental or bacterial agents from the gastrointestinal tract. These specific physiological conditions cause continuous antigen exposure and require an intrinsic immunosuppressive environment to prevent autoimmune damage [20]. Additionally, etiologic factors of HCC, such as chronic viral infection and other inflammatory liver disorders, increase the expression of PD-L1, which is associated with a higher tolerance toward tumor-associated antigens and favorable conditions for HCC tumorigenesis, as well as the recurrence of the primary tumor after surgical resection [21,22].

It is well known today that the immune system plays an important role in controlling cancer progression [23]; several immune mechanisms are important in the development and progression of HCC and correlate with prognosis [24]. Both innate and adaptive immune systems work to enable effective anticancer immune surveillance, and a dysfunctional tumor-immune system interaction leads to immune evasion through impaired antigen recognition or by generating an immunosuppressive tumor microenvironment (TME) [25]. Several molecular alterations are known to contribute to an immunosuppressive TME, including the presence of regulatory T cells, inhibitory B cells, myeloid-derived suppressor cells and/or M2-polarized tumor-associated macrophages. In addition, the upregulation of coinhibitory lymphocyte signals, such as immune checkpoint ligands and receptors, the elevated levels of tolerogenic enzymes, including indoleamine 2,3-dioxygenase-1 or arginase-1, and also reduced immunoglobulin-mediated opsonization, are all factors related to an immunosuppressive TME [26]. ICIs are monoclonal antibodies that inhibit the deactivation of T lymphocytes by blocking the interaction of checkpoint proteins with their ligands. They have demonstrated that an efficient immune response can get rid of tumor cells in a way that greatly enhances cancer treatment outcomes [25].

### 2.3. Molecular and Immune HCC Classes

A better understanding of the histologic and molecular landscape of HCC has helped us identify and classify different HCC subgroups based on specific histologic and molecular alterations, and this classification can help guide treatment decisions in the near future [26]. Alcohol-related HCCs are significantly enriched in CTNNB1, TERT, CDKN2A, SMARCA2 and HGF alterations. Hepatitis B virus (HBV)-related HCCs are frequently mutated in TP53. By contrast, hepatitis C virus (HCV) infection, metabolic syndrome and hemochromatosis usually do not show significant associations [19].

Patients with HCC can be placed into two groups based on the molecular features: the proliferative or non-proliferative class. The proliferative class presents a higher prevalence of HBV infections, is associated with high levels of α-fetoprotein (AFP), presents a worse prognosis and is characterized by the activation of PI3K–AKT–mTOR, RAS–MAPK and MET signaling along with chromosomal instability [27]. In this scenario, tumor progression involves the production of VEGF, which promotes the vascularization and angiogenesis of malignant tissue. Therefore, treatments based on antiangiogenic profiles, such as kinase inhibitors, are terapeuticoptions for these patients. At present, the best available first-line treatment for advanced HCC is a combination of a PDL1 blockade with atezolizumab and a VEGF blockade with bevacizumab [24]. The second group, the non-proliferation HCC class, includes cases with a history of alcohol abuse and HCV infection, with a better prognosis, and is characterized by mutations in CTNNB1, the gene encoding β-catenin [27].

Regarding immune classes, approximately one-third of tumors fall into the “inflamed class”, which is characterized by high levels of immune cells in the tumor microenvironment, high cytolytic activity, increased levels of PD1 and PD-L1 expression, activated interferon signaling and low chromosomal alterations. These tumors are considered “hot tumors” and include a subgroup with high interferon signaling and CTNNB1 mutations. It is believed that tumors in this category tend to have an increased likelihood of responding to immune checkpoint inhibitors, although this finding is yet to be defined and is currently being evaluated. By contrast, “cold” tumors have little T-cell presence and are either characterized by TP53 mutations (intermediate class) or the activation of WNT signaling through CTNNB1 mutations (excluded class) [28,29].

## 3. First-Line Treatment

The treatment of advanced HCC has evolved significantly in the past decade with the introduction of novel target agents. Trials conducted after 1980 on systemic chemotherapy, hormonal therapy, interferon and combination regimens showed overall response rates (ORRs) of 0–28%. However, this improvement came at the cost of significant toxicities, and in subsequent randomized controlled studies, those treatments failed to show benefits in terms of OS [30]. Therefore, these regimens are of limited value in clinical practice.

New treatment regimens with tyrosine kinase inhibitors, immune checkpoint inhibitors and combinations are described in the text below and summarized in Table 1.

### 3.1. Tyrosine Kinase Inhibitors

With the introduction of sorafenib, a new standard of care was adopted. This targeted therapy was the first to show efficacy in patients with advanced HCC based on the results of two phase III trials: SHARP and Asia-Pacific [10,11]. These studies showed significant OS improvement in patients who received sorafenib treatment.

After ten years, another targeted therapy, lenvatinib, was evaluated in the first-line treatment of patients with unresectable HCC. This oral multikinase inhibitor targets VEGF receptors 1–3 and fibroblast growth factor receptors (FGFR1-4), PDGFRα and KIT, and rearranged during transfection (RET) [44]. The approval for lenvatinib treatment was based on the results of the phase III randomized noninferiority trial REFLECT. This study compared lenvatinib given at 12 mg once daily (for bodyweight ≥ 60 kg) or 8 mg daily (for bodyweight < 60 kg) versus sorafenib given at 400 mg twice daily in 28-day cycles. The trial included 954 patients, 99% of whom were in Child-Pugh class A and had never received systemic treatment. The involvement of more than 50% of the liver or an invasion of the main portal vein or biliary tree were major exclusion criteria. Those treated with lenvatinib experienced a median OS of 13.6 months, compared to patients treated with sorafenib, who had a median OS of 12.3 months (HR 0.92; 95% CI: 0.79–1.06), meeting the predetermined noninferiority margin of 1.08. The experimental group reported a median progression-free survival (PFS) of 7.4 months while the control group had a median PFS of 3.7 months (HR: 0.66; *p* < 0.0001). Lenvatinib therapy had an ORR of 24.1% compared to 9.2% for the sorafenib arm (OR 3.13; *p <* 0.0001). Grade three or higher treatment-related adverse events (TRAEs) were documented in 57% of patients in the experimental group and 49% of patients in the control group. Hypertension (23%) and weight loss (8%) were the grade three or higher TRAEs most frequently observed in the lenvatinib arm [31].

### 3.2. Immune Checkpoint Inhibitors and Combination Regimens

Despite improvements in OS, treatment protocols with multikinase inhibitors alone failed to show robust response rates until the development of immune checkpoint inhibitors (ICIs) and their evaluation as a systemic treatment option for HCC. This class of drugs targets specific subtypes of membrane-bound molecules that act as pivotal regulators of immune escape in cancer [18,24]. The monoclonal antibodies classified as ICIs have two main targets: cytotoxic T lymphocyte protein 4 (CTLA-4) and the pair composed of programmed cell death protein-1 and its ligand (PD-1, PD-L1) [45].

In 2020, the phase III randomized trial IMbrave150 reported its results comparing the combinations of the ICI atezolizumab and anti-VEGF bevacizumab against sorafenib in patients with locally advanced metastatic or unresectable HCC. This trial randomized 501 patients who were assigned to receive 1200 mg of atezolizumab plus 15 mg per kilogram of body weight of bevacizumab or sorafenib at 400 mg orally twice daily. The updated median OS of the atezolizumab-bevacizumab group was 19.2 months versus 13.4 months in the sorafenib group (HR 0.66; 95% CI: 0.52–0.85; *p* < 0.001). The median PFS was 6.9 months in the experimental group versus 4.3 months in the control group (HR 0.65; 95% CI: 0.53–0.81; *p* < 0.001). The confirmed ORRs were 30% (95% CI, 25–35) with atezolizumab-bevacizumab and 11% (95% CI, 7–17) with sorafenib. Twenty-five patients (8%) in the experimental group had complete responses (CR) versus one (<1%) in the control group. The study was interrupted prematurely, having met its primary endpoint of OS at the first interim analysis [32,33]. The safety analysis reported similar results regarding all-cause adverse events (98% in the atezolizumab-bevacizumab group versus 99% in the sorafenib group) and increased numbers of serious adverse events in the experimental group (49% versus 33%), including six grade five bleeding incidents [32]. Because of the increased risk of gastrointestinal bleeding associated with bevacizumab, screening for esophageal varices is recommended before initiating treatment. Despite this result, the IMbrave 150 trial successfully demonstrated the potential of ICIs to treat advanced HCC and become a standard of care according to global guidelines [46,47,48,49].

Another combination tested with atezolizumab was the multikinase inhibitor cabozantinib in the COSMIC-321 phase III randomized trial. This study evaluated atezolizumab-cabozantinib versus sorafenib (with a dual primary endpoint of PFS and OS) and sorafenib versus the cabozantinib single agent (with a secondary endpoint of PFS). Eight hundred and thirty-seven patients with advanced HCC were randomized to receive cabozantinib tablets at 40 mg orally once daily plus atezolizumab at 1200 mg intravenously every three weeks, sorafenib 400 mg orally twice daily or single agent cabozantinib tablets at 60 mg orally once daily. Despite improving the median PFS from 4.2 months in the sorafenib group to 6.8 months in the combination group (HR 0.63; 99% CI: 0.44–0.91, *p* = 0.0012), the study did not demonstrate significant benefits regarding OS at the interim analysis. The median reported OS was 15.4 months in the experimental group versus 15.5 months in the control group (HR: 0.90; 96% CI 0.69–1.18; *p* = 0·44). Single-agent cabozantinib showed a median PFS of 5.8 months versus 4.3 months for sorafenib at the interim analysis (HR: 0.71; 99% CI: 0.51–1.01, *p* = 0.011). The safety data were consistent with previously reported toxicities for cabozantinib, sorafenib and atezolizumab [34].

The combination of anti-PD-L1 durvalumab plus anti-CTLA 4 tremelimumab showed encouraging results in the HIMALAYA trial. The managers of this large phase III study recruited 1171 patients and initially had four arms comparing two different regimens of durvalumab-tremelimumab or durvalumab as a single agent, both compared with sorafenib. After the discontinuation of one of the combination regimens due to the poor results of a phase 1/2 study [50], the remaining arms included a single dose application of 300 mg of tremelimumab + durvalumab 1.500 mg every four weeks (Single Tremelimumab Regular Interval Durvalumab—STRIDE regimen), durvalumab 1.500 mg every four weeks and sorafenib 400 mg twice daily. The reported results showed a median OS in the STRIDE regimen of 16.43 months versus 13.77 months in the sorafenib group (HR: 0.78; 96.02; CI: 0.65–0.93; *p* = 0.0035). Furthermore, the single agent durvalumab showed noninferiority to sorafenib when compared with the prespecified noninferiority margin of 1.08 (HR: 0.86; 95.67 CI: 0.73–1.03). The ORRs were 20.1% and 17% for STRIDE and single-agent durvalumab, respectively, versus 5.1% for sorafenib. However, despite the OS and ORR results, neither experimental regimen significantly extended the PFS, which was 3.78 months for STRIDE (HR 0.90; 95% CI: 0.77–1.05; *p* = 0.0035), 3.65 months for durvalumab alone (HR 1.02; 95% CI: 0.88–1.19; *p* = 0.0674) and 4.07 months for sorafenib. Regarding the toxicity profile, the incidence of TRAEs was lower in the experimental regimens (75.8% for STRIDE, 52.1% for durvalumab alone) when compared with sorafenib (84.8%). Grade 3/4 immune-mediated events were slightly more common in the STRIDE group (25.6%) versus 36.9% in the sorafenib group and 12.9% in the durvalumab single-agent group. Based on those results, the FDA approved the STRIDE regimen as an option for the first-line treatment of unresectable HCC [35].

Another ICI that demonstrated efficacy in this setting was the anti-PD 1 tislelizumab. The noninferiority RATIONALE-301 trial randomized 674 patients to receive 200 mg of tislelizumab every three weeks or 400 mg of sorafenib twice daily. In the final analysis of the study, patients treated with tislelizumab had a median OS of 15.9 months versus 14.1 months in the sorafenib group (HR: 0.85; 95.003% CI: 0.712–1.019), achieving the noninferiority threshold of 1.08. Despite improvement in ORR (14.3% with tislelizumab versus 5.4% with sorafenib), the PFS was not improved with the experimental treatment (2.2 months with tislelizumab versus 3.6 months with sorafenib; HR: 1.1; 95% CI 0.92–1.33). Incidence rates of grade three or higher adverse effects were 48.2% for the experimental treatment versus 65.4% for the control group. The most common immune-mediated adverse effects were hepatitis (5.5%) and hypothyroidism (5.3%) [36].

The phase III randomized CheckMate 459 trial evaluated nivolumab at a dose of 240 mg every two weeks versus sorafenib at 400 mg twice daily as a first-line treatment for advanced HCC. The study randomized 743 patients and demonstrated an ORR of 15%, with 4% being CRs versus 7% on the control arm. However, statistically significant results were not observed for OS, with a median OS of 16.4 months with nivolumab versus 14.7 with sorafenib (HR: 0.85; 95% CI: 0.72–1.02; *p* = 0.0752). Grade three or higher TRAEs were less frequent in nivolumab-treated patients (22%) than in sorafenib-treated patients (49%) [37].

In cohort B of the phase I/II study on CheckMate 040, patients with advanced HCC and Child-Pugh B cirrhosis were treated with nivolumab at 240 mg every two weeks. The trial included 25 patients who were previously untreated and 24 patients who were previously treated with sorafenib. With 16.3 months of median follow-up, the ORR was 12% (95% CI 5–25) with a disease control rate (DCR) of 55% (95% CI 40–69). The median duration of response (DOR) was 9.9 months (95% CI 9.7–9.9), and the median OS was 9.8 months (95% CI 3.7–14.3) for sorafenib-naïve patients and 7.4 months (95% CI 2.3–12.1) for previously treated patients. Disease progression was the most common reason for treatment discontinuation (78%). Grade three or higher TRAEs were reported in 51% of the patients, and the most common were hypertransaminasemia (4%) and increased amylase (4%) [38].

The lenvatinib plus pembrolizumab combination was evaluated in the phase Ib, open-label multicenter trial KEYNOTE-524. This study enrolled 104 patients who were treated in the first-line setting with lenvatinib 12 mg daily if their body weight was >60 kg and 8 mg if their body weight was <60 kg, plus pembrolizumab at 200 mg every three weeks in a single arm. The ORR per modified Response Evaluation Criteria in Solid Tumors (mRECIST) was 46% (95% CI: 36.0–56.3%) with 11% CR, and per RECIST v1.1, the ORR was 36% (95% CI: 26.6–46.2%) with 1% CR. After 10.6 months of median follow-up, the median PFS was 9.3 months (95% CI: 5.6–9.7 months) per mRECIST and 8.6 months (95% CI: 7.1−9.7 months) per RECIST v1.1. The median OS was 22.0 months (95% CI: 20.4−NR months) per mRECIST criteria. Grade three or higher TRAEs were reported in 67% of patients. Hypertension was the most common grade three TRAE (17%), followed by AST increase (11%) and diarrhea (5%) [39].

The combination of lenvatinb plus pembrolizumab was also evaluated in the multicenter phase III study LEAP-002. This trial evaluates levantinib plus pembrolizumab versus levantinib in the first-line treatment of advanced HCC using the same protocols as KEYNOTE-524. The study randomized 794 patients and had dual endpoints of OS and PFS. The reported median OS in the final analysis was 21.2 months in the combination group and 19 months for the lenvatinib group (HR: 0.84; 95% CI: 0.708–0.997; *p* = 0.0227; superiority threshold, one-sided alfa = 0.0185). The median PSF in the final analysis was 8.2 months for the experimental group and 8.1 months for the control group (HR: 0.834; 95% CI: 0.712–0.978). The combination had an ORR of 26.1% per RECIST v1.1 (40.8% per mRECIST); lenvatinib had an ORR of 17.5% per RECIST v1.1 (34.1% per mRECIST). Safety results were consistent with earlier studies with hypertension as the most common TRAE (43.3% in the combination group and 46.8% in the control group), followed by diarrhea (40.3% in the combinantion group and 33.9% in the control group) and hypothyroidism (40% in the combination group and 35.7% in the control group). After the final analysis, the study did not meet the pre-specified statistical significance for the primary endpoints of OS and PFS [41].

The anti-PD-1 IgG4 camrelizumab and the VEGFR2-targeted TKI rivoceranib were tested in combination in a phase III trial as a first-line therapy for unresectable HCC. The study randomized 543 patients to either camrelizumab 200 mg every two weeks plus rivoceranib 250 mg daily or sorafenib 400 mg daily. Reported median PFS in ITT population were 5.6 months for the experimental therapy and 3.7 for the control group (HR: 0.62; 95% CI: 0.49–0.80; *p* < 0.0001). The median OS were 22.1 months for the combination therapy and 15.2 months for sorafenib (HR: 0.62; 95% CI: 0.49–0.80; *p* < 0.0001). The ORR per RECIST v1.1 was 25.4% (95% CI: 20.3–31%) for the experimental group and 5.9% (95 CI: 3.4–9.4%). The overall response rate per mRECIST was 33.1% (95% CI: 27.5–39%) for the combination and 10% (95% CI: 6.7–14.2) for the control group. Safety analysis showed 80.5% of grade 3-4 TRAEs for the combination arm and 52% for the sorafenib arm, and both sides had one occurrence of grade five TRAE. The most common side effect was hypertension (69% for the experimental group and 37.5% for the control group), followed by AST increase (54% for the experimental group and 16.5% for the control group) and proteinuria (49.3% for the experimental group and 5.9% for the control group) [42].

### 3.3. Other Treatment Options under Investigation

The human liver has a dual blood supply. Healthy hepatic cells mainly receive oxygenated blood from the portal vein, while liver tumors, alternatively, are supplied by the hepatic artery [51]. Despite significant improvements in the systemic therapy of HCC already discussed, there are other methods of first-line treatments that exploit these anatomical characteristics.

Interventional hepatic arterial infusion chemotherapy (HAIC) is a therapeutic technique that directly delivers agents into tumor-associated arterial branches using a catheter implanted by surgery or interventional radiology. The potential of directly infusing chemotherapy on the hepatic arteries includes increased antitumor activity by bypassing first passage, resulting in higher dose of medication delivered in the liver, and lower systemic toxicity than standard intravenous therapy. This treatment was already explored in non-randomized studies and in a phase II trial, all of them including mainly Asiatic patients, with clinical benefit [52,53,54].

The Chinese phase III study FOHAIC-1 compared hepatic arterial infusion chemotherapy of Oxaliplatin plus Fluorouracil versus sorafenib. Two hundred and sixty-two patients with locally advanced or unresectable HCC were randomly assigned to FOLFOX (oxaliplatin 130 mg/m^2^, leucovorin 200 mg/m^2^, fluorouracil 400 mg/m^2^ and infusional fluorouracil 2400 mg/m^2^ every three weeks or sorafenib at 400 mg twice daily. Of note, a significant percentage of the study population presented macrovascular invasion (including portal vein invasion Vp-4), tumor involvement > 50% of the liver or extrahepatic oligometastasis. The majority of patients in the study (89%) had HBV-related HCC. At the median follow-up of 17.1 months, the median OS was 13.9 months in the HAIC group and 8.2 months in the sorafenib group (HR: 0.408; *p* < 0.001). The subgroup of patients with high risk factors (defined as Vp4 portal vein tumor thrombosis and/or >50% liver occupation) had a median OS of 10.8 months in the experimental group and 5.7 months in the control group (HR 0.343; 95% CI: 0.219–0.538). Median PFS was 7.8 months in the experimental group and 4.3 months in the control group (HR: 0.451; *p* < 0.001). Disease downstage occurred in 16 patients (12.3%) in the HAIC group and 15 patients received curative or locoregional treatment. The ORR according to RECIST was 31.5% in the experimental group compared to 1.5% in the sorafenib group (mRECIST: 35.4% vs. 5.3%; *p* < 0.001). TRAES were recorded in 100% of the sorafenib population and in 94.5% of the HAIC population. The most common grade 3-4 TRAEs in the HAIC group were elevated AST (10.9%) and thrombocytopenia (10.9%). Catheter-related adverse events were observed in six (4.7%) patients [40].

The transarterial chemoembolization (TACE) is another form of treatment that explores the peculiarities of hepatic tumor vascularity. TACE consists in an infusion of particles that, at the same time, deliver targeted chemotherapy and promote the embolization of arteries supplying the tumor. The results are the regression of localized lesions as well as the up-regulation of proangiogenic growth factors [55].

The phase III trial LAUNCH evaluated the combination of TACE and lenvatinib in primary treatment-naive or initial recurrent advanced HCC after surgery. The study proposed a synergistic antitumor effect of the combination via the use of a multikinase inhibitor to block pro-angiogenic factors elevated after TACE. The study randomized 338 patients in China. Key inclusion criteria were advanced HCC that was treatment-naïve or recurrent after a radical resection without adjuvant treatment and Child-Pugh class A. Eligible patients who had only a single intrahepatic lesion (≤10.0 cm) or multiple lesions (≤10 foci) were admitted if the tumor burden was <50%. Patients received lenvatinib given at 12 mg once daily (for bodyweight ≥ 60 kg) or 8 mg daily (for bodyweight < 60 kg) combined with TACE (LEN-TACE) or lenvatinib alone. Again, the majority of the study population (86.3%) had HBV-related HCC. The median OS was 17.8 months in the LEN-TACE group and 11.5 months in the lenvatinib group (HR: 0.33; *p* < 0.001). The median PFS was 10.6 months in the experimental group and 6.4 months in the control group (HR: 0.36; *p* < 0.001). Grade 3-4 TRAE were more common in the LEN-TACE group and included AST elevation (22.9%) and hyperbilirubinemia (9.4%). The most common TRAE in the experimental group was hypertension (64%), followed by abdominal pain (50.6%) and diarrhea (47.1%) [43].

## 4. Second-Line and Beyond

The emergence of immune checkpoint inhibitors has revolutionized HCC treatment. Currently, there is no standard second-line treatment after progression to immunotherapy. For patients treated with sorafenib or lenvatinib in the first-line, there are five options approved by the Food and Drug Administration (FDA): three antiangiogenics (regorafenib, ramucirumab and cabozantinib), a combination of immunotherapy (nivolumab and ipilimumab) and pembrolizumab.

Trials evaluating treatments in second-line and beyond scenarios are described in the text above and summarized in Table 2.

### 4.1. Tyrosine Kinase Inhibitors

In the RESORCE study, 573 patients who had failed sorafenib treatment but who had tolerated treatment well and maintained ECOG 0-1 and Child-Pugh A were randomized in a 2:1 ratio to regorafenib 160 mg or a placebo once daily during weeks 1–3 of each four-week cycle. Regorafenib improved OS with a HR of 0.63 (95% CI 0.50–0.79; one-sided *p* < 0·0001); the median OS was 10.6 months (95% CI: 9.1–12.1) for regorafenib versus 7.8 months (95% CI: 6.3–8.8) for the placebo. Adverse events were reported in all regorafenib recipients and 179 (93%) of the 193 placebo recipients. The most common clinically relevant grade three or four TRAEs were hypertension (15% in the regorafenib group versus 5% in the placebo group), hand-foot skin reaction (13% versus 1%), fatigue (9% versus 5%) and diarrhea (3% versus no patients) [56].

In the CELESTIAL trial, 707 Child-Pugh A patients who had failed any line of treatment were randomized to cabozantinib (60 mg daily) or a placebo. There were 495 subjects treated with prior sorafenib in this group. The results showed a gain in OS (11.3 versus 7.2 months; HR = 0.7; *p* = 0.005) but a low response rate (4 versus 1%). Cabozantinib improved PFS compared with the placebo irrespective of the duration of prior sorafenib. The median PFS was 3.8 for cabozantinib versus 1.8 months for the placebo (HR 0.35, 95% CI 0.23 to 0.52) for patients who received sorafenib. The benefit was maintained even in patients who had used sorafenib for less than three months. The most common grade 3/4 TRAEs in the cabozantinib arm were palmar-plantar erythrodysesthesia (16%), hypertension (16%) and increased aspartate aminotransferase (12%) [57].

In the REACH trial, 565 patients who had failed prior treatment with sorafenib and remained Child-Pugh A were randomized to ramucirumab, at 8 mg/kg every two weeks or a placebo. In the intention-to-treat population, the use of ramucirumab did not result in a significant gain in OS (9.2 versus 7.6 months; HR = 0.87; *p* = 0.14). Grade three or greater TRAEs occurring in 5% or more of patients in either treatment group were ascites (5% for ramucirumab arm versus 4% in the placebo arm), hypertension (12% versus 4%), asthenia (5% versus 2%), increased aspartate aminotransferase concentration (5% versus 8%), thrombocytopenia (5% versus <1%) and hyperbilirubinemia (1% versus 5%). In this study, a subgroup analysis of patients with alpha-fetoprotein (AFP) > 400 ng/mL (250 subjects) indicated a potential benefit to OS with ramucirumab in this population (7.8 versus 4.2 months; HR = 0.67; *p* = 0.0059) [58]. Based on these subgroup data, the REACH-2 study included 292 patients with AFP > 400 ng/mL who were randomized to receive 8 mg/kg intravenous ramucirumab every two weeks or a placebo. At a median follow-up of 7.6 months, the median OS was 8.5 months versus 7.3 months (HR = 0.710; *p* = 0.0199), and the median PFS was 2.8 months versus 1.6 months (HR: 0.452; *p* < 0.0001). Both endpoints were significantly improved in the ramucirumab group compared with the placebo group. The proportion of patients with an objective response did not differ significantly between groups (5% versus 1; *p* = 0.1697) [59].

Brivanib is a selective dual inhibitor of VEGF and the fibroblast growth factor receptors implicated in the tumorigenesis and angiogenesis of HCC. The randomized phase III BRISK-PS study assessed brivanib in patients with HCC who had been treated with sorafenib. The study randomized 395 patients who were previously treated with or were intolerant to sorafenib. The experimental arm was best supportive care (BSC) and brivanib 800 mg orally once per day and the control arm was placebo plus BSC. Brivanib did not substantially enhance OS, as seen by the median OS (the primary endpoint) values of 9.4 months for brivanib and 8.2 months for the placebo (HR: 0.89; *p* = 0.3307). The median PFS was 4.2 months for brivanib and 2.7 months for the placebo (HR: 0.56; *p* < 0.001), and the mRECIST ORR was 10% for brivanib and 2% for the placebo (odds ratio, 5.72). Brivanib’s grade three to four TRAE with the highest frequency were hypertension (17%), fatigue (13%), hyponatremia (11%) and reduced appetite (10%) [60].

A meta-analysis by the Mayo Clinic group suggests that among second-line options after sorafenib, there was greater benefit in OS with regorafenib, cabozantinib or ramucirumab (only if AFP > 400). This study included five trials in the second-line analysis encompassing a total of 2653 patients who were involved in evaluating five drugs (cabozantinib, regorafenib, ramucirumab, brivanib and pembrolizumab). In comparison to the placebo, the meta-analysis revealed that all medications improved PFS. However, only cabozantinib (HR, 0.76; 95% CI, 0.63–0.92) and regorafenib (HR, 0.62; 95% CI, 0.51–0.75) showed improved OS compared to the placebo. Regorafenib, cabozantinib and ramucirumab all outperformed the placebo in the subgroup of patients with AFP levels of 400 ng/mL or above in terms of PFS and OS [60].

Apatinib, an oral VEGFR-2 inhibitor, was investigated in a phase III randomized placebo-controlled study. This trial evaluated 393 patients with advanced HCC and Child-Pugh A or B (≤7) cirrhosis following the failure of sorafenib and oxaliplatin-based chemotherapy. Patients got either a placebo or 750 mg of apatinib orally once each day. Overall survival was the primary endpoint. Both the median OS (8.7 months in the apatinib arm compared to 6.8 months in the placebo arm, HR 0.785; *p* = 0.0476) and median PFS (4.5 months with apatinib compared to 1.9 months with a placebo, HR 0.471; *p* < 0.0001) were significantly extended by apatinib. Compared to the placebo group, the apatinib group had an ORR of 10.7% (95% CI: 7.2–15.1%) as opposed to 1.5% (95% CI: 0.2–5.4%). The most common grade three or four TRAEs were hypertension (28% in the apatinib group versus 2% in the placebo group), hand-foot syndrome (18% versus none) and decreased platelet count (13% versus 1%) [61].

Although lenvatinib has never been prospectively evaluated in the second-line after using anti-PD-L1 plus VEGF inhibitor, a retrospective study of 48 patients demonstrated a higher response rate (15.8 versus 0%) with lenvatinib (12 mg once daily (for bodyweight ≥ 60 kg) or 8 mg daily (for bodyweight < 60 kg)) versus sorafenib (400 mg twice daily). In this study, patients were treated with sorafenib (*n* = 29), lenvatinib (*n* = 19) and cabozantinib (*n* = 1). In all patients, the ORR and disease control rate were 6.1 and 63.3%, respectively. The median PFS and OS for all patients were 3.4 months (95% CI: 1.8–4.9) and 14.7 months (95% CI: 8.1–21.2), respectively. Lenvatinib had a considerably longer median PFS than sorafenib (6.1 vs. 2.5 months; *p* = 0.004), but there was no difference in the median OS (16.6 vs. 11.2 months; *p* = 0.347). Forty-two patients (85.7%) had TRAEs of any degree, and eight patients (16.3%) had TRAEs of grade three [62].

### 4.2. Immune Checkpoint Inhibitors

The phase I/II study CheckMate 040 enrolled patients with HCC in different cohorts. Patients were allocated in three different arms. Patients in arm A received nivolumab 1 mg/kg plus ipilimumab 3 mg/kg administered every three weeks (for four cycles), followed by nivolumab 240 mg every two weeks. Arm B received nivolumab 3 mg/kg plus ipilimumab 1 mg/kg administered every three weeks (for four cycles), followed by nivolumab 240 mg every two weeks, and arm C received nivolumab 3 mg/kg every two weeks plus ipilimumab 1 mg/kg every six weeks [62].

In the randomized, phase III trial on Keynote 240, 413 Child-Pugh A HCC patients were selected after progression or intolerance to sorafenib and were randomized to 200 mg of pembrolizumab or saline placebo intravenously every three weeks for at least 35 cycles (approximately two years). Pembrolizumab resulted in a higher rate of response (18.3% versus 4.4%) (*p* = 0.0174 in the final analysis). The median OS was 13.9 months for pembrolizumab versus 10.6 months for the placebo (HR = 0.781; *p* = 0.0238). The median PFS for pembrolizumab was 3.0 months versus 2.8 months at the final analysis (HR, 0.718; *p* = 0.0022). Grade three or higher TRAEs that occurred more frequently with pembrolizumab than the placebo were increased AST level (13.3% versus 7.5%), increased blood bilirubin level (7.5% versus seven 5.2%) and increased ALT (6.1% versus 3.0%). In this study, the OS and PFS did not reach statistical significance per specified criteria [63].

Keynote-394, a randomized, double-blind, phase III study was conducted in Asia to evaluate the efficacy and safety of pembrolizumab versus a placebo, both of which were given with the best support care as second-line therapy for previously treated advanced HCC. A total of 453 patients were randomized to pembrolizumab 200 mg once every three weeks for ≤ 35 cycles (N = 300) or a placebo (N = 153) plus the best supportive care. The primary endpoint was OS, and the secondary endpoints included PFS and ORR. Pembrolizumab showed a significantly longer median OS of 14.6 months versus 13.0 months of the placebo arm (HR = 0.79; *p* = 0.0180) and a longer median PFS of 2.6 versus 2.3 months (HR = 0.74; *p* = 0.0032). Additionally, the ORR was 12.7% for pembrolizumab versus 1.3% for placebo (*p* < 0.0001). TRAEs were noted in 66.9% of patients in the pembrolizumab arm and in 49.7% of patients in the placebo group [64].

### 4.3. Other Immune Checkpoint Inhibitors under Investigation

In the second-line scenario, more ICIs are being investigated. In a phase I/II study, 332 patients with advanced HCC, who had progressed on or were intolerant to sorafenib, were randomly assigned to four different arms. One of them involved administering a combination of tremelimumab 300 mg plus durvalumab 1500 mg (T300 + D) at one dose each during the first cycle, followed by durvalumab 1500 mg once every four weeks. The other arms consisted in durvalumab monotherapy 1500 mg once every four weeks, tremelimumab monotherapy (750 mg once every four weeks for a total of seven cycles and then every 12 weeks) and the combination of tremelimumab 75 mg once every four weeks plus durvalumab 1500 mg once every four weeks for four cycles, followed by durvalumab 1500 mg every four weeks (T75 + D). The ORRs were 24.0% (95% CI: 14.9 to 35.3), 10.6% (95% CI: 5.4 to 18.1), 7.2% (95% CI: 2.4 to 16.1) and 9.5% (95% CI: 4.2 to 17.9), respectively. The median OS was 18.7 months (95% CI: 10.8 to 27.3) for T300 + D arm, 13.6 month (95% CI: 8.7 to 17.6) for durvalumab monotherapy, 15.1 months (95% CI: 11.3 to 20.5) for tremelimumab monotherapy and 11.3 months (95% CI: 8.4 to 15.0) for T75 + D combination [65].

Aptatinib is being tested in conjunction with the anti-PD-1 antibody camrelizumab in patients with advanced HCC, gastric cancer (GC) or esophagogastric junction cancer (EGJC). The results of a phase I study, including dosage escalation and expansion cohorts, were published. A total of 43 patients (18 with advanced HCC and 25 with GC/EGJC) were recruited in the research. Among HCC patients, fifteen patients had disease progression or were intolerant to sorafenib and 16 were considered evaluable. Partial response (PR) was observed in eight patients (50%) and seven patients (43%) had stable disease (SD) as best response. The median PFS of HCC patients was 5.8 months (95% CI: 2.6–not reached) and the median OS was not reached. In phase Ia part of the study, four dose-limiting toxicities were noted (26.7%), including three grade three pneumonitis events in the apatinib 500 mg group and one grade three lipase increase in the apatinib 250 mg cohort. Based on the results of this trial, the recommended phase II dosage (RP2D) for apatinib was 250 mg [66].

Camrelizumab (at a dose of 3 mg/kg intravenously every two to three weeks) was evaluated in patients with advanced HCC who had progressed on or were intolerable to prior systemic therapy in a multicenter, open-label, randomized, phase II trial conducted in China. This trial randomized 220 eligible patients, and 217 of them got camrelizumab (109 patients were given treatment every two weeks and 108 every three weeks). The ORR, which was the primary endpoint, was 14.7% (95% CI: 10.3–20.2). The six month OS rate was 74.4% (95% CI 68.0–79.7). Aspartate aminotransferase elevation (5%) and reduced neutrophil count (3%) were the most prevalent TRAEs [67].

Another anti-PD-1 monoclonal antibody, cemiplimab, is being investigated as a potential therapy for patients with advanced HCC as a second-line treatment. A phase I study with an expanded cohort evaluated patients with advanced HCC who had progressed after receiving first-line therapy. The trial included 26 patients. Five patients had a PR (ORR of 19.2%) as best response, whereas 14 patients (53.8%) had SD. The median PFS was 3.7 months (95% CI: 2.3−9.1). The most common TRAEs of any grade were fatigue (26.9%), decreased appetite, increased aspartate aminotransferase (AST), abdominal pain, pruritus and dyspnea (each 23.1%) [68].

Tislelizumab is an investigational, humanized, IgG4 monoclonal antibody that targets programmed cell death-1 (PD-1) with high specificity and affinity. It has been designed to reduce binding to FcγR on macrophages in order to avoid antibody-dependent phagocytosis, a process that can clear T-cells and lead to potential resistance to anti-PD-1 therapy. A phase I/ II study conducted in 16 centers in China included patients ≥ 18 years old with histologically confirmed advanced/metastatic solid tumors, including HCC. The patients must have progressed since their last standard antitumor treatment, had no available (or refused) standard treatment or become intolerant to treatment and have adequate organ function. The ORR in the HCC cohort was 17%. Across all patients in the study (*n* = 300), the median OS was 11.5 months (95% CI 9.1 to 15.0), with a median follow-up of 12.2 months. The OS data remained immature for HCC. The median PFS for all patients was 2.6 months (95% CI: 2.2 to 4.0), and for HCC patients, it was 4.0 months. Among the patients in the trial who had responses, most experienced durable decreases in their tumor burden. These durable responses were observed in all indications, even in patients who were heavily pretreated. During the dose-verification portion of the trial, the RP2D was confirmed to be tislelizumab 200 mg intravenously every three weeks. The most common reported TRAEs were anemia (*n* = 104; 35%), increased aspartate aminotransferase (*n* = 75; 25%) and increased alanine aminotransferase (*n* = 67; 22%) [69].

## 5. Sequencing Treatment

Fortunately, more patients have been exposed to systemic therapies and can receive not just one line but two or three lines of therapy. Atezolizumab combined with bevacizumab, along with durvalumab and tremelimumab, has become the preferred first-line option for those without contra-indications to immunotherapy. However, bevacizumab should be avoided in those at high risk of bleeding, such as those with untreatable esophageal varices (Figure 1).

Unfortunately, the available prospective data on the effectiveness of second-line immunotherapy has only been collected from patients initially treated with sorafenib. There is no information from phase III trials on how well it works in patients initially treated with bevacizumab and atezolizumab. Additionally, there is no data to guide the best order for later treatment options after any initial regimen [70]. In the absence of such data, the side effect profile of each regimen must be considered carefully, and eligible patients should participate in clinical trials whenever possible. If clinical trials are not feasible, alternative treatments to atezolizumab with bevacizumab or durvalumab/tremelimumab include sorafenib, lenvatinib, regorafenib and cabozantinib (Figure 1).

Despite the absence of prospective data, a retrospective study evaluated the efficacy of cabozantib as second-line treatment in patients with advanced HCC who progressed on immunotherapy. This study collected data from patients who were treated with cabozantinib between 2010 and 2021 at Mayo Clinic locations in Minnesota, Arizona and Florida. Twenty-six patients were evaluated. The median OS after starting cabozantinib treatment was 7.7 months (95% CI: 5.3–14.9), and the median PFS was 2.1 months (95% CI: 1.3–3.9). The ORR was 4%. Despite its retrospective nature, this study shows that individuals who have progressed on immunotherapy may benefit from cabozantinib therapy as second-line treatment [71].

For patients initially treated with sorafenib or lenvatinib (and without contra-indications to immunotherapy), the authors recommend using immune checkpoint inhibitors (such as nivolumab with ipilimumab or pembrolizumab alone) as second-line therapy over tyrosine kinase inhibitors. This recommendation is based on data suggesting a higher response rate and a more favorable side effect profile. However, it is important to note the absence of trials with head-to-head comparisons in this setting. Situations like this can happen when the choice of sorafenib or lenvatinib in the first-line was made not due to contra-indications to immunotherapy, but to the unavailability of atezolizumab, bevacizumab, durvalumab and tremelimumab.

## 6. Biomarkers

Despite extensive research and genomic analyses that have improved our molecular understanding of HCC, no potential biomarker has been identified to predict responses or guide targeted therapies [19,72]. Some druggable targets, such as gene fusions involving NTRK, are extremely rare in HCC [73].

In posterior analyses of archived tumor tissues and baseline plasma samples from patients with HCC in the RESORCE trial, some expression patterns of plasma proteins and miRNAs were associated with major benefits, and increased the median OS following treatment with regorafenib. Five of the 266 analyzed proteins were identified as predictive of regorafenib treatment benefit for OS (angiopoietin 1—ANG1 [HR: 1.12; *p*: 0.019], cystatin B [HR: 1.46; *p* = 0.040], the latency-associated peptide of transforming growth factor beta 1—LAP TGF-b1 [HR: 1.36; *p* = 0.040], oxidized low-density lipoprotein receptor 1—LOX-1 [HR: 1.35; *p* = 0.009] and C-C motif chemokine ligand 3—MIP-1a [HR: 1.02; *p* = 0.040]). Increased miRNA plasma levels of MIR30A (HR: 1.47; *p* = 0.003), MIR122 (HR: 1.35; *p* = 0.0004), MIR125B (HR: 1.54; *p* = 0.001), MIR200A (HR: 1.39; *p* = 0.001) and MIR374B (HR: 1.36; *p* = 0.002) decreased the miRNA levels of MIR15B (HR: 0.37; *p* = 0.002), MIR107 (HR: 0.54; *p* = 0.003) and MIR320B (HR: 0.57; *p* = 0.001), and the absence of MIR645 (HR: 3.16; *p* = 0.002) were all predictive of a survival benefit with regorafenib. MIR15B, MIR320B and MIR200A were also prognostic for OS (*p* ≤ 0.05) [74].

BIOSTORM was a study designed to define predictors of recurrence prevention with sorafenib and prognosis after hepatectomy. The study analyzed tumor tissue from 188 patients from the STORM trial. Gene expression profiling, targeted exome sequencing (19 known oncodrivers), immunohistochemistry (pERK, pVEGFR2 and Ki67), fluorescence in situ hybridization (VEGFA) and the immunome were investigated. While hepatocytic pERK (HR: 2.41; *p* = 0.012) and microvascular invasion (HR: 2.09; *p* = 0.017) were independent prognostic factors, no mutation, gene amplification or previously proposed gene signatures predicted the sorafenib benefit. A novel 146-gene expression signature was associated with improved recurrence-free survival (RFS) with adjuvant sorafenib treatment after hepatectomy and had a statistically significant predictive value (*p* = 0.04). The patients identified as ‘sorafenib RFS responders’ did not reach the median RFS with sorafenib, while the ‘non-responders’ median RFS was 28 months [75].

Plasma levels of AFP and c-MET are prognostic biomarkers associated with poor outcomes [74]. Except for AFP, which is a predictive biomarker for the ramucirumab survival benefit [76], no other biomarker is used in clinical practice to guide therapeutic decisions involving target drugs in HCC.

### 6.1. Biomarkers for Immunotherapy

No single robust biomarker has been identified thus far for immunotherapy in HCC. Some have been studied and might be considered, although they are not required in clinical practice.

#### 6.1.1. PD-L1 Expression

PD-1/PD-L1 checkpoint inhibitors are largely the most popular immunotherapy drugs. They are antibodies against the membrane receptors PD-1 and PD-L1, which are involved in controlling T-cell migration, proliferation and the secretion of cytotoxic mediators [77].

In a recent analysis to explore biomarkers of the CheckMate 040 trial, high PD-L1 expression on tumor cells was associated with improved survival (mOS 28.1 months [95% CI 18.2–n.a.] for patients with PD-L1 ≥ 1% versus 16.6 months [95% CI 14.2–20.2] for patients with PD-L1 < 1%; *p* = 0.032), and high PD-1 expression in tumors was associated with an improved ORR (*p* = 0.05). In the same study, the IHC assessment of tumor-infiltrating T-cell expression demonstrated that higher densities of CD3+ or CD8+ tumor-infiltrating lymphocytes (TILs) tended toward improved OS (both *p* = 0.08) [78]. Similarly, an expression of PD-L1 ≥ 1% was found to be associated with a longer median OS in those treated with nivolumab versus sorafenib (16. versus 8.6 months; HR: 0.80; 95% CI: 0.54–1.17) in the CheckMate 459 study [37].

Sorafenib exposure appeared to change the PD-L1 expression and gene signatures within the tumor microenvironment in another study with tislelizumab, an anti-PD-1 monoclonal antibody. Sorafenib use was associated with higher PD-L1 expression (PD-L1+ prevalence 53.7% versus 25%; *p* = 0.08) and fewer immune-suppressive signatures, which were implicated in response, and PFS from tislelizumab. In sorafenib-exposed patients, the ORR was higher in PD-L1+ patients than in PD-L1 patients (ORR 23.8% versus 0%; *p* = 0.049) [79].

In the Keynote 224 trial, the response to pembrolizumab seemed to be linked to PD-L1 expression quantified with expression in both tumor cells and nontumor cells lymphocytes and macrophages for CPS (ORR 32% versus 20%; *p* = 0.021), but not PD-L1 expression on tumor cells alone, and TPS (ORR 43% versus 22%; *p* = 0.088) [80].

In the recent phase III HIMALAYA trial, the PD-L1 status, measured by CPS, was not shown to be linked with the benefit of doublet immunotherapy over that of sorafenib in overall survival (PD-L1 expression positive: HR 0.85–95% CI 0.65–1.11 vs. PD-L1 expression negative: HR 0.83–95% CI 0.65–1.05) [81].

Instead of PD-L1 testing by immunohistochemistry, which has been approved by the FDA as an acceptable biomarker for several cancers, a lack of standard methods for evaluating PD-L1 expression and its spatial and temporal heterogeneity still limits its use in addressing HCC [82].

#### 6.1.2. Tumor Mutational Burden and Microsatellite Instability

Cancer neoantigens arise as a consequence of tumor-specific mutations and are major factors in the activity of clinical immunotherapies [83]. The tumor mutational burden (TMB) is defined as the total number of mutations per coding area of a tumor genome and is a biomarker of response to anti-PD-1 therapy [84,85]. A deficiency in mismatch repair (dMMR) activity results in a hypermutator phenotype known as microsatellite instability (MSI), which contains exceptionally high numbers of somatic mutations [86] and is considered an agnostic histological indicator for the selection of responders to ICI therapy [87]. High TMB or high MSI are found in less than 2% of HCC cases, which limits their exploration or use as biomarkers [88].

#### 6.1.3. Other Possible Biomarkers: Circulating Tumor Cells, Gut Microbiota and WNT/β-Catenin Signaling

A liquid biopsy comprises the analysis of circulating tumor components, such as circulating tumor cells (CTCs), DNA and extracellular vesicles. It may enable early cancer detection, the prediction of treatment response and the molecular monitoring of the disease [89]. In a pivotal study, Winograd et al. reported that PD-L1+ CTCs were a prognostic biomarker of worse OS. Among ten patients with HCC receiving immune checkpoint inhibitors (nivolumab or pembrolizumab), only patients who had PD-L1+ CTCs seemed to have any response (all five responders demonstrated PD-L1+ CTCs at baseline, compared with only one of five non-responders) [90].

The human intestinal tract is inhabited by a large number of microorganisms, including bacteria, viruses and fungi known as the gut microbiota, which is the largest microbiota in the body. The gut microbiota interacts with the body, participating in digestion and metabolism, and is closely related to the immune state of the body. These commensal microorganisms could be used as emerging biomarkers in immunotherapy and may support the immunotherapy efficacy or lack thereof in treating various cancer types [91]. Higher taxonomic diversity and more gene counts in 20 species, including *Akkermansia* and *Ruminococcaceae* in fecal samples, were associated with a better response to ICIs and might be an early response biomarker [92].

The lack of T-cell infiltration and resistance to anti-PD-L1/anti-CTLA-4 monoclonal antibody therapy was correlated with altered β-catenin signaling in melanoma [93]. In patients with HCC treated with ICIs, the presence of activating WNT/β-catenin mutations was also associated with a lower disease control rate (0% vs. 53%), shorter median PFS (2.0 vs. 7.4 months) and shorter median OS (9.1 vs. 15.2 months) [94].

## 7. Conclusions

The development of new therapeutic options has improved the management of HCC over the past decade. As illustrated in this article, ICIs or target agents alone have a relatively low response rate. Thus, the combination of different ICIs or the combination of immunotherapy with other treatments, such as targeted and locoregional therapies, are promising strategies to treat HCC, and combined therapy is already the current standard of care for first-line treatment [95]. Currently, some randomized phase III clinical trials testing combinations of treatment modalities are being conducted in advanced HCC patients (Table 3). As an example, which is supported by CheckMate 040 [62], the phase III CheckMate-9DW trial (NCT04039607) is comparing nivolumab plus ipilimumab with the physician’s choice of sorafenib or lenvatinib in the first-line treatment of patients with advanced HCC.

As we have observed in this manuscript, immunotherapy plays a fundamental role in the management of advanced HCC, and new, sophisticated immune therapies such as CAR-T or CAR-NK cells have been developed and may expand the armamentarium for HCC in the long term [96,97]. Currently, there are at least five registered pivotal studies with CAR-T cells targeting glypican 3 (GPC3) or allogenic NK cells for Barcelona C HCC patients (NCT04121273; NCT03198546; NCT02905188; NCT04106167; and NCT03841110).

Transforming growth factor-β (TGF-β) seems to have opposite roles as a carcinogen [98]. It is a tumor suppressor during the early phase and promotes cancer development during the late phase by inducing epithelial-to-mesenchymal transition and downregulating antitumor immunity [99]. Targeting the TGF-β pathway is a promising strategy for cancer therapy and has shown promising results when combined with sorafenib [100]. In addition, the TGFβ pathway inhibitor galunisertib regulates T-cell immunity and has synergistic antitumor effects with PD-1/L1 inhibitors [101]. Galunisertib plus nivolumab for HCC is being evaluated in a phase I/II trial (NCT02423343).

Improvements in the understanding of the molecular landscape have the potential to guide treatment advances with the development of new targeted agents with improved clinical activity. Fibroblast growth factor 19 (FGF19) and fibroblast growth factor receptor 4 (FGFR4) are critically involved in the development of HCC by inhibiting apoptosis and promoting proliferation and invasion [102]. At present, FGFR4-selective inhibitors are being tested for HCC alone (NCT02834780) or associated with anti-PD-L1 mAb (NCT04194801). Additionally, in a xenograft model of HCC, the coadministration of the cyclin-dependent kinase 4/6 inhibitor palbociclib in combination with FGFR4-selective inhibitors facilitated tumor regression, which indicates a potential new strategy for HCC treatment [103].

Lastly, the subclassification of HCC based on distinct molecular and histologic subtypes will soon allow us to personalize treatment for advanced HCC to select the best treatment for the best candidate based on molecular subtypes and biomarkers with the potential to markedly increase survival outcomes and hopefully cure some patients.

## Figures and Tables

**Figure 1 cancers-15-01680-f001:**
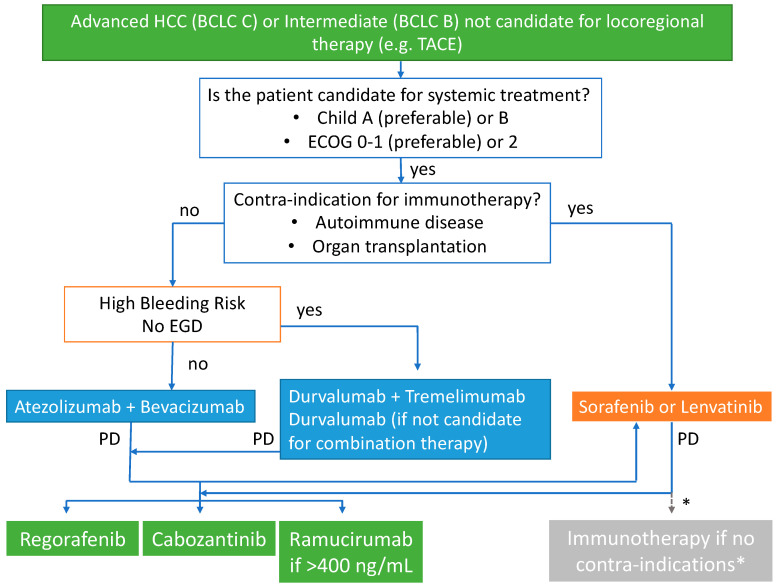
Proposed sequencing algorithm for advanced HCC.

**Table 1 cancers-15-01680-t001:** Results of selected trials for first-line therapy in patients with advanced HCC.

Study (Year)	Phase	N	Population	Geographical Region	Drug	Median Overall Survival	Median Progression-Free Survival	Objective Response Rate
REFLECT trial (2018) [31]	III noninferiority	954	Unresectable HCC and no prior systemic therapy (99% Child-Turcotte-Pugh class A)	29% (white); 69% (Asian); 2% (other)	Lenvatinib vs. sorafenib	13.6 mo for lenvatinib vs. 12.3 mo for sorafenib (HR: 0.92, 95% CI: 0.79–1.06)	7.4 mo for lenvatinib vs. 3.7 mo for sorafenib (HR: 0.66; *p* < 0.0001)	24.1% for lenvatinib vs. 9.2% for sorafenib (*p* < 0.0001)
IMbrave 150 trial (2021) [32,33]	III	336	Unresectable or metastatic HCC, Child-Pugh liver function score < 7, and no prior systemic therapy	40% (Asians, excluding Japan); 60% (rest of the world)	Atezolizumab-bevacizumab vs. sorafenib	19.2 mo for atezolizumab-bevacizumab vs. 13.4 mo for sorafenib (HR: 0.66; *p* < 0.001)	6.9 mo for atezolizumab-bevacizumab vs. 4.3 mo for sorafenib (HR: 0.65; *p* < 0.001)	30% to atezolizumab-bevacizumab vs. 11% to sorafenib
COSMIC-321 trial (2022) [34]	III	837	Unresectable or metastatic HCC, Child-Pugh liver function score < 7, and no prior systemic therapy	29.3% (Asians); 70.7% (Other)	Cabozantinib-atezolizumab vs. sorafenib	15.4 mo for cabozantinib-atezolizumab vs. 15.5 mo for sorafenib (HR: 0.90, *p* = 0.44)	6.8 mo for cabozantinib-atezolizumab vs. 4.2 mo for sorafenib (HR: 0.63, *p* = 0.0012)	13% to cabozantinib-atezolizumab vs. 6% to sorafenib
HIMALAYA trial (2022) [35]	III	1171	Unresectable HCC, Child-Pugh liver function score < 7, and no prior systemic therapy	40.9% (Asians, excluding Japan); 59.1% (rest of the world)	Durvalumab-tremelimumab or durvalumab vs. sorafenib	16.43 mo for STRIDE vs. 13.77 for sorafenib (HR: 0.78; *p* = 0.0035)	3.78 mo for STRIDE and 3.65 mo for durvalumab vs. 4.07 for sorafenib (HR: 0.90; *p* = 0.0035 and HR: 1.02, *p* = 0.0674)	20.1% to STRIDE, 17% to durvalumab vs. 5.1 to sorafenib
RATIONALE-301 trial (2022) [36]	III	674	Unresectable or metastatic HCC, Child-Pugh liver function score < 7, and no prior systemic therapy	63.1% (Asians, excluding Japan); 11.4 (Japan); 25.5% (rest of the world)	Tislelizumab vs. sorafenib	15.9 mo for tislelizumab vs. 14.1 mo for sorafenib (HR: 0.8)	2.2 mo for tislelizumab vs. 3.6 mo for sorafenib (HR: 1.1)	14.3% to tislelizumab vs. 5.4% to sorafenib
CheckMate 459 (2019) [37]	III	743	Unresectable Child-Pugh A HCC naïve to systemic treatment	40% (Asian); 60% (United States, Canada or Europe)	Nivolumab vs. sorafenib	16.4 mo for nivolumab vs. 14.7 mo for sorafenib (HR: 0.85; *p* = 0.0752)	3.7 mo for nivolumab vs. 3.8 mo for sorafenib	15% for nivolumab and 7% to sorafenib
CheckMate-040: cohort B (2021) [38]	I/II	49	Unresectable or metastatic HCC, Child-Pugh liver function score B, with or without prior systemic therapy	55% (Asian); 41% (white); 2% (black); 2% (other)	Nivolumab single arm	9.8 mo for sorafenib naïve patients and 7.4 mo for previously treated patients	3.4 mo for sorafenib naïve patients and 2.2 mo for previously treated patients	12%
KEYNOTE-524 trial (2022) [39]	Ib	104	Unresectable or metastatic HCC, Child-Pugh liver function score < 7, and no prior systemic therapy	51% (white); 28% (Asian); 2% (black); 5% (other); 14% (missing)	Lenvatinib-pembrolizumab single arm	22 mo	9.3 mo per mRECIST; 8.6 per RECIST v1.1	46% per mRECIST; 36% per RECIST v1.1
FOHAIC-1 (2021) [40]	III	262	Locally advanced or unresectable HCC with or without extrahepatic oligometastasis, Child-Pugh liver function score ≤ 7		HAIC (FOLFOX) vs. sorafenib	13.9 mo for HAIC vs. 8.2 mo for sorafenib (HR: 0.408, *p* < 0.001)	7.8 mo for HAIC vs. 4.3 mo for sorafenib (HR: 0.451, *p* < 0.001)	31.5% to HAIC and 1.5% to sorafenib per RECIST; 35.4% to HAIC and 5.3% to sorafenib per mRECIST (*p* < 0.001)
LEAP-002 (2021) [41]	III	794	Primary treatment-naive HCC, non-amenable to curative therapy, Child-Pugh A	30.7% (Asian without Japan) vs. 69.3% (western regions and Japan)	lenvantinib plus pembrolizumab vs. lenvantinib	21.2 mo for lenvatinib and pembrolizumab vs. 19 mo for lenvatinib (HR: 0.84; *p* = 0.0227)	8.2 mo for lenvatinib and pembrolizumab vs. 8.1 mo for lenvatinib (HR: 0.834; *p* = 0.0466)	26.1% for lenvatinib and pembrolizumab and 17.5% for lenvatinib per RECIST 1.1; 40.6% for lenvatinib and pembrolizumab and 34.1% for lenvatinib per mRECIST
Qin, et al. (2022) [42]	III	543	Unresectable or metastatic HCC primary treatment naive, BCLC satage B, Child-Pugh A	82.7% (Asian) vs. 17.3% (non-Asian)	Camrelizumab + rivoceranib vs. sorafenib	22.1 mo for canrelizumab + rivoceranib vs. 15.2 mo for sorafenib (HR: 0.62; 95% CI: 0.49–0.80)	5.6 mo for canrelizumab + rivoceranib vs. 3.7 mo for sorafenib (HR: 0.52; 95% CI: 0.41–0.65)	25.4% for camrelizumab and rivoceranib and 5.9% for sorafenib per RECIST 1.1; 33.1% for camrelizumab and rivoceranib and 10% for sorafenib per mRECIST
LAUNCH (2022) [43]		338	Primary treatment-naive or initial recurrent advanced HCC after surgery without adjuvant treatment, Child-Pugh class A	100% (Asian—China)	LEN-TACE vs. lenvatinib	17.8 mo forLEN-TACE vs. 11.5 mo for lenvatinib (HR: 0.33; *p* < 0.001)	10.6 mo forLEN-TACE vs. 6.4 mo for lenvatinib (HR: 0.36; *p* < 0.001)	45.9% to LEN_TACE and 20.8% to lenvatinib per RECIST; 54.1% to LEN-TACE and 25% to lenvatinib per mRECIST (*p* < 0.001)

Abbreviations: HCC: hepatocellular carcinoma; mo: months; HR: hazard ratio; RECIST: Response Evaluation Criteria in Solid Tumors; vs.: versus.

**Table 2 cancers-15-01680-t002:** Results of selected trials in second-line and beyond for patients with advanced HCC.

Study (Year)	Phase	N	Population	Drug	Median Overall Survival	Median Progression-Free Survival	Objective Response Rate
RESORCE trial (2017) [56]	III	573	Advanced HCC that progressed after first-line treatment with sorafenib, Child-Pugh A	Regorafenib vs. placebo	10.6 mo for regorafenib vs. 7.8 mo for placebo (HR: 0.63; *p* < 0.0001)	3.1 mo for regorafenib vs. 1.5 mo for placebo (HR: 0.46; *p* < 0.0001)	11% for regorafenib vs. 4% for placebo (*p* = 0.0047)
CELESTIAL trial (2018) [57]	III	707	Advanced and progressing HCC and not worse than Child-Pugh A	Cabozantinib vs. placebo	10.2 mo for cabozantinib vs. 8.0 mo for placebo (HR: 0.76; *p* = 0.005)	5.2 mo for cabozantinib vs. 1.9 mo for placebo (HR: 0.44; *p* < 0.001)	4% for cabozantinib vs. less than 1% for placebo (*p* = 0.009)
REACH trial (2015) [58]	III	565	Advanced HCC following first-line therapy with sorafenib and Child-Pugh A	Ramucirumab vs. placebo	9.2 mo for ramucirumab vs. 7.6 mo for placebo (HR: 0.87; *p* = 0.14).	2.8 mo for ramucirumab vs. 2.1 mo for placebo (HR 0.63; *p* < 0.0001)	7% for ramucirumab vs. < 1% for placebo (*p* < 0.0001)
REACH-2 trial (2019) [59]	III	292	Advanced HCC, Child-Pugh class A, and serum AFP ≥ 400 ng/mL in patients who had disease progression under first-line sorafenib	Ramucirumab vs. placebo	8.5 mo for ramucirumab vs. 7.3 mo for placebo (HR: 0.71; *p* = 0.0199	2.8 mo for ramucirumab vs. 1.6 mo for placebo (HR: 0.452; *p* < 0. 0001)	5% for ramucirumab vs. 1% for placebo (*p* = 0.1697)
BRISK-PS study [60]	III	395	Advanced HCC who progressed on/after or were intolerant to sorafenib	Brivanib vs. placebo	9.4 mo for brivanib vs. 8.2 mo for placebo (HR: 0.89; *p* = 3307)	4.2 mo for brivanib vs. 2.7 mo for placebo (HR, 0.56; *p* < 0.001)	10% for brivanib vs. 2% for placebo (odds ratio, 5.72).
Qiu Li et al. (2020) [61]	III	393	Advanced HCC after failure of sorafenib and oxaliplatin-based chemotherapy and Child-Pugh liver function class A or B ≤ 7 points	Apatinib vs. placebo	8.7 mo for apatinib vs. 6.8 mo for placebo (HR: 0.785; *p* = 0.0476)	4.5 mo for apatinib vs. 1.9 mo for placebo (HR: 0.471; *p* < 0.0001)	10.7% for ramucirumab vs. 1.5% for placebo
CheckMate 040 (2020) [62]	I/II	148	Advanced HCC patients who were treatment-naive or received sorafenib previously	Nivolumab and ipilimumab	Arm A: 22.8 mo Arm B: 12.5 mo ArmC: 12.7 mo		Arm A: 32% Arm B: 27% Arm C: 29%
KeyNote 240 (2019) [63]	III	413	Child-Pugh A HCC patients, after progression or intolerance to sorafenib	Pembrolizumab vs. BSC	NR	13.8 months	18.3 versus 4.4%.
KeyNote 394 (2022) [64]	III	453	Second-line therapy for previously treated advanced HCC	Pembrolizumab vs. placebo	14.6 vs. 13.0 months	2.6 vs. 2.3 months	12.7% vs. 1.3%,
Kelley et al.. (2020) [65]	I/II	332	Advanced HCC patients who progressed on, were intolerant to or refused sorafenib	Durvalumab + tremelimumab	T300 + D: 18.7 mo Durvalumab: 13.6 mo Tremelimumab: 15.1 mo T75 + D: 11.3 mo	T300 + D: 2.17 mo Durvalumab: 2.07 mo Tremelimumab: 2.69 mo T75 + D: 1.87 mo	T300 + D: 24.0% Durvalumab: 106% Tremelimumab: 7.2% T75 + D: 9.5%
Xu et al. (2019) [66]	I	18	HCC patients, Child-Pugh A, and previously treated with sorafenib	SHR-1210 + apatinib	NR	5.8 mo	50%
Qin et al. (2020) [67]	II	217	Advanced HCC, Child-Pugh A or B7 after sorafenib failure or intolerance to first-line systemic therapy	Camrelizumab	13.8 mo	2.1 mo	14.7%
He et al. (2018) [68]	Ib	26	Advanced HCC, Child-Pugh A after failure or intolerance to first-line systemic therapy	Cemiplimab		3.7 mo	19.2%
Shen et al. (2020) [69]	I/II	300	Advanced or metastatic solid tumors, including HCC, in patients who have progressed since their last standard antitumor treatment, had no available (or refused) standard treatment, or become intolerant to treatment	Tislelizumab	immature for HCC	4.0 mo	

Abbreviations: HCC: hepatocellular carcinoma; mo: months; HR: hazard ratio; NR: not reached; HBV: hepatitis B virus; HCV: hepatitis C virus; vs.: versus.

**Table 3 cancers-15-01680-t003:** Ongoing phase III trials in patients with advanced HCC.

Drugs	Phase	Setting	Endpoint	ClinicalTrials.gov Identifier
SBRT followed by sintilimab vs. SBRT	II/III	Palliative—1st line	Primary: 24-week PFS rate Secondary: PFS; OS; ORR; DCR; DOR	NCT04167293
Lenvatinib + pembrolizumab +TACE vs. TACE	III	Palliative—1st line	Primary: PFS; OS Secondary: ORR; DCR; DOR; TTP; AEs	NCT04246177
Arm A: TACE + durvalumab; Arm B: TACE + durvalumab + bevacizumab; Arm C: TACE	III	Palliative—1st line	Primary: PFS (Arm B vs. Arm C) Secondary: PFS (Arm A vs. Arm C); OS	NCT03778957
Nivolumab + Ipilimumab vs. sorafenib or lenvatinib	III	Palliative—1st line	Primary: OS Secondary: ORR; DOR; TTSD	NCT04039607
Finotonlimab (anti PD1) + SCT510 (bavacizumab) vs. Sorafenib	II/III	Palliative—1st line	Primary: OS Secondary: ORR; PFS	NCT04560894
Toripalimab + Lenvatinib vs. Lenvatinib	III	Palliative—1st line	Primary: OS, PFS Secondary: ORR; DOR; TTP	NCT04523493
Nofazinlimab (CS1003) + Lenvatinib vs. Lenvatinib	III	Palliative—1st line	Primary: OS, PFS	NCT04194775
Atezolizumab + Lenvatinib or Sorafenib vs. Lenvatinib or Sorafenib	III	Palliative—2nd line	Primary: OS Secondary: ORR; PFS; DOR; TTP	NCT04770896

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
