# Peer review of "Systemic Therapy for Advanced Hepatocellular Carcinoma: Current Stand and Perspectives"

_cancers, 2023, doi:10.3390/cancers15061680_

Round 1

Reviewer 1 Report

It is a review of current systemic therapies for HCC. There have been a lot of changes recently and authors have tried to summarize the results. Please see below comments to improve this review

1) On page 3, second paragraph: authors mention that 26% have actionable mutation based on a reference. I would be cautious about these statements as These include alterations where FDA approved drugs have minimal efficacy. That is one of the reason that its pretty rare to have HCC patients treated with targeted drug used for specific alterations in real world setting. Secondly, its difficult with liver dysfunction which is frequently associated in this patient population

2) Table 1: median os for atezo+ bev is incorrect

3) Table 1: some trials are missing: LEAP-02 trial

4) Table 1: I would suggest another column that includes proportion of asian patients vs western

5) Table 1: camrelizumab plus rivoceranib phase 3 trial should be added

6) Page 7, para 2: Authors mention that cosmic-312 didnt show OS benefit due to subsequent therapies. Not sure if it is true. In HCC, there is not always a correlation between PFS and OS. Is there a data on subsequent therapies that author know that be included before making the conclusion

7) Keynote-240 should be mentioned and included in table

8) page 15, last para: this small study could be mention for data on cabozantinib post IO ( PMID: 36358592)

9) page 17: 3rd para: HImalaya study didnt show difference with PDl1 expression and should be included 

Author Response

Dear Reviewer 1

I think we have successfully addressed all the issues noted and we feel that the manuscript is significantly improved. Above is a point-by-point response to your comments.

It is a review of current systemic therapies for HCC. There have been a lot of changes recently and authors have tried to summarize the results. Please see below comments to improve this review

  • On page 3, second paragraph: authors mention that 26% have actionable mutation based on a reference. I would be cautious about these statements as These include alterations where FDA approved drugs have minimal efficacy. That is one of the reason that its pretty rare to have HCC patients treated with targeted drug used for specific alterations in real world setting. Secondly, its difficult with liver dysfunction which is frequently associated in this patient population

Thank you for your comments. We agree with the reviewer and have decided to delete this paragraph from the manuscript.

  • Table 1: median os for atezo+ bev is incorrect

Thank you for noticing. Updated median OS was corrected.

  • Table 1: some trials are missing: LEAP-02 trial

Thank you. Leap02 trial was added in the table and in the manuscript

  • Table 1: I would suggest another column that includes proportion of asian patients vs western

Thank you. A new column was included according to your suggestion.

  • Table 1: camrelizumab plus rivoceranib phase 3 trial should be added

Thank you. This trial was added in the table and in the manuscript

  • Page 7, para 2: Authors mention that cosmic-312 didnt show OS benefit due to subsequent therapies. Not sure if it is true. In HCC, there is not always a correlation between PFS and OS. Is there a data on subsequent therapies that author know that be included before making the conclusion

Thank you. We agree with your comment and excluded this sentence from the manuscript

  • Keynote-240 should be mentioned and included in table

The Keynote 240 trial was already described in the manuscript and was already described in table 2 – line 8. We kept it both in the manuscript and in the table

  • page 15, last para: this small study could be mention for data on cabozantinib post IO ( PMID: 36358592)

Thank you. A paragraph about this study was included in the manuscript

  • page 17: 3rd para: HImalaya study didnt show difference with PDl1 expression and should be included 

Thank you. We included a paragraph with this data in the manuscript.

Reviewer 2 Report

The manuscript is well described regarding the current treatment options for the advanced HCC.

Minor points,

1. The authors recommend using ICI as 2nd-line over TKI in Figure 1 and lines 577-582, although the patients using TKI (e.g. Sorafenib or Lenvatinib) as 1st-line are categorized as "contra-indication for immunotherapy". This sequencing would not be appropriate.

2. In lines 450-451, the authors describe "after using a VEGF inhibitor". But, I recommend "after using anti-PD-L1 plus VEGF inhibitor" to request accuracy, because this study was conducted to evaluate the efficacy of Lenvatinib/Sorafenib/Cabozantinib after Atezolizumab plus Bevacizumab. 

Author Response

Dear Reviewer

I think we have successfully addressed all the issues noted and we feel that the manuscript is significantly improved. Above is a point-by-point respons to your comments.

  1. The authors recommend using ICI as 2nd-line over TKI in Figure 1 and lines 577-582, although the patients using TKI (e.g. Sorafenib or Lenvatinib) as 1st-line are categorized as "contra-indication for immunotherapy". This sequencing would not be appropriate.

We agree with the reviewer. That was the reason why the arrows pointing at those options (IO after sorafenib/levantinib) were in different colors and, in the text, we stated the following: “and without contra-indications to immunotherapy”. Unfortunately, situations like this can happen when the choice of sorafenib or lenvatinib in the first-line was made not due to contraindications to immunotherapy, but to the unavailability of atezolizumab, bevacizumab, durvalumab and tremelimumab. To better clarify this topic, we have added a sentence to the paragraph explaining that this situation might happen in clinical practice and added modifications to the figure.2.

In lines 450-451, the authors describe "after using a VEGF inhibitor". But, I recommend "after using anti-PD-L1 plus VEGF inhibitor" to request accuracy, because this study was conducted to evaluate the efficacy of Lenvatinib/Sorafenib/Cabozantinib after Atezolizumab plus Bevacizumab. 

Thank you. We corrected this sentence according to your suggestion.